# Early Cardiac Dysfunction in Duchenne Muscular Dystrophy: A Case Report and Literature Update

**DOI:** 10.3390/ijms26041685

**Published:** 2025-02-16

**Authors:** Maria Lupu, Iustina Mihaela Pintilie, Raluca Ioana Teleanu, Georgiana Gabriela Marin, Oana Aurelia Vladâcenco, Emilia Maria Severin

**Affiliations:** 1Clinical Neurosciences Department, Paediatric Neurology, Faculty of Medicine, Carol Davila University of Medicine and Pharmacy, 020021 Bucharest, Romania; maria.lupu@rez.umfcd.ro (M.L.); raluca.teleanu@umfcd.ro (R.I.T.); oana-aurelia.vladacenco@drd.umfcd.ro (O.A.V.); emilia.severin@umfcd.ro (E.M.S.); 2Department of Paediatric Neurology, Dr. Victor Gomoiu Children’s Hospital, 022102 Bucharest, Romania; 3Clinical Cardiology Department of Oncological Institute, Prof. Dr. Alexandru Trestioreanu, 022328 Bucharest, Romania; georgiana-gabriela.marin@rez.umfcd.ro

**Keywords:** Duchenne muscular dystrophy, dilated cardiomyopathy, genotype–phenotype, dystrophin gene, exon skipping, cardiac pathology

## Abstract

Duchenne Muscular Dystrophy (DMD) is a severe X-linked recessive disorder characterized by progressive muscle degeneration due to dystrophin deficiency. Cardiac involvement, particularly dilated cardiomyopathy, significantly impacts morbidity and mortality, typically manifesting after age 10. This case report presents a rare instance of early-onset cardiac involvement in a 3-year-old male with a confirmed deletion in exon 55 of the dystrophin gene. The patient developed dilated cardiomyopathy at 3 years and 8 months, with progressive left ventricular dysfunction despite early treatment with corticosteroids, ACE inhibitors, and beta-blockers. Genetic mechanisms and genotype–phenotype correlations related to cardiac involvement were reviewed, highlighting emerging therapies such as exon skipping, vamorolone, ifetroban, and rimeporide. Studies indicate that variants in exons 12, 14–17, 31–42, 45, and 48–49 are associated with more severe cardiac impairment. This case emphasizes the need for early, ongoing cardiac assessment and personalized treatment to address disease heterogeneity. While current DMD care standards improve survival, optimizing management through early intervention and novel therapies remains essential. Further research is needed to better understand genotype–phenotype correlations and improve cardiac outcomes for patients with DMD.

## 1. Introduction

Duchenne muscular dystrophy (DMD), (OMIM #310200; ORPHA:98896), an X-linked dystrophinopathy, manifests predominantly in males. The global prevalence of DMD was recorded at 7.1 cases per 100,000 males and 2.8 cases per 100,000 in the general population [1]. The incidence of DMD is 1 in 3500 male newborns [2].

As the most severe form of muscular dystrophy, DMD results from dystrophin deficiency, which typically manifests in early childhood [3]. The condition follows an X-linked recessive inheritance pattern, with initial clinical signs usually emerging between ages 3 and 5, including muscle weakness, fatigue, and gait abnormalities due to proximal muscle involvement [4]. Over time, the disease leads to the loss of ambulation and a significantly reduced life expectancy, often into the third or fourth decade of life [5,6].

Dystrophin is an essential protein that links the dystrophin–glycoprotein complex on the muscle membrane to actin filaments inside the cell, enabling force transmission from muscle contraction to the extracellular matrix [7,8]. When dystrophin is absent or dysfunctional, the muscle membrane becomes fragile, leading to increased cellular permeability, which is reflected by elevated levels of creatine kinase (CK) and cardiac troponin (cTn) in the blood [9].

Cardiac involvement in DMD typically manifests as dilated cardiomyopathy (DCM) and arrhythmias. Clinical signs of cardiomyopathy usually appear after the age of 10, with approximately one-third of patients affected by age 14 and all patients showing signs of heart disease by age 18 [10]. Since muscle weakness precedes cardiac symptoms, early diagnosis and management of cardiovascular complications are critical for improving survival rates and quality of life [10,11,12].

We present a case of a patient diagnosed with DMD, who exhibited early-onset dilated cardiomyopathy at just 3 years and 8 months of age. This report underscores the clinical course, multidisciplinary management, and the potential influence of genetic factors on cardiac involvement. Notably, the patient’s genetic variant, a deletion in exon 55 of the *DMD* gene, has been previously reported in the literature, but without phenotypic data. This highlights the need for further genotype–phenotype correlation studies to better understand the implications of specific genetic variants in DMD [13]. Additionally, we review the current literature to provide insights into emerging therapeutic strategies for DMD-related cardiac dysfunction.

## 2. Detailed Case Description

### 2.1. Methodology

We present a case of a 19-year-old male patient with DMD and early-onset cardiac involvement. Retrospective data collection included all available clinical, paraclinical, and multidisciplinary assessments. The patient was evaluated and monitored in accordance with international standards of care, with regular follow-ups from a multidisciplinary team.

Cardiac function was assessed through routine echocardiographic and electrocardiographic monitoring. Genetic testing was performed using single-gene filtered Next-Generation Sequencing (NGS) on a peripheral blood sample, identifying a pathogenic variant in the *DMD* gene. This variant was classified as pathogenic according to the American College of Medical Genetics and Genomics (ACMG) criteria [14].

This study was approved by the Ethics Committee of Dr. Victor Gomoiu Children’s Hospital, and informed consent was obtained from the patient’s legal guardians.

### 2.2. Clinical Presentation and Diagnosis

The patient had no notable family history and was born following an unremarkable pregnancy and an uneventful neonatal period. He presented motor delay; at the age of one year and 6 months, he was not walking independently, and the laboratory results at that time revealed elevated levels of liver transaminases and CK of 12,300 IU. Additionally, abdominal ultrasound showed hepatomegaly. The patient was referred to the pediatric neurology service, where suspicion of a neuromuscular disorder was raised following clinical and paraclinical assessments.

### 2.3. Genetic Testing

Given the patient’s clinical presentation, characterized by motor delay and significantly elevated CK levels suggestive of muscular dystrophy, we performed single gene-filtered NGS of the *DMD* gene. This revealed a pathogenic variant in exon 55 (NM_004006.3, DMD: g.1716784_1716785del; c.8064_8065delTA; p.(His2688GlnfsTer21)), classified as pathogenic according to the American College of Medical Genetics and Genomics (ACMG) criteria (PVS1—Pathogenic Very Strong 1, PM2—Pathogenic Moderate 2, PS4—Pathogenic Strong 4), thus confirming the diagnosis of Duchenne muscular dystrophy at the age of three [15]. This genetic finding is illustrated in Figure 1. Our literature review identified a single article reporting the same genetic variant found in our patient. However, that study did not include any phenotypic data, limiting the ability to establish genotype–phenotype correlations [13].

### 2.4. Multidisciplinary Team and Initial Findings

Due to the multiple comorbidities associated with DMD, the patient received care in accordance with international standards through the coordinated efforts of a multidisciplinary team. Neurological examination revealed several significant findings: the patient walked with a wide gait and had difficulty walking on the heels. Additionally, he presented with flat feet, pseudohypertrophy of the calf muscles, motor deficits in both the shoulder and pelvic girdle, a positive Gower’s sign, decreased myotatic reflexes, and dyslalia. A physical therapist performed an initial assessment and developed a customized exercise program focusing on maintaining muscle strength, range of motion, and functional independence. The patient and family were educated about the importance of regular physical activity and adherence to the prescribed exercise regimen. The NSAA (North Star Ambulatory Assessment) scale indicated a score of 30 out of 30 points, while the 6 min walk (6MWT) test demonstrated a distance of 375 m. Baseline pulmonary function tests were performed to assess respiratory function, with no pathological changes detected. Respiratory assessments revealed no signs of breathing difficulties, as all measured parameters remained within normal ranges, indicating no requirement for supplementary respiratory support such as non-invasive ventilation. A gastroenterologist performed a comprehensive nutritional assessment and developed a nutrition plan to ensure adequate calorie and nutrient intake. Counseling and support services were provided to help the patient and family cope with the emotional impact of the diagnosis. From a psychological standpoint, the patient exhibited an IQ (intelligence quotient) of 80. Psychiatric evaluation revealed a diagnosis of hyperkinetic disorder with attention deficit, the predominantly inattentive type. The patient was enrolled in endocrinological monitoring, and at that time, corticosteroid treatment had not yet been initiated. The patient underwent orthopedic monitoring, during which no pathological findings were detected.

### 2.5. Cardiac Monitoring and Follow-Ups

Initially, no significant changes in cardiac function were observed. However, during a routine evaluation at the age of 3 years and 8 months, a grade I of IV systolic murmur was identified. This finding prompted a detailed cardiological investigation, which revealed signs consistent with cardiomyopathy. Despite the absence of symptoms, the echocardiographic examination confirmed longitudinal dysfunction of the left ventricle (LV) with an ejection fraction (EF) of 56.69%, along with evidence of mild ventricular dilation, as seen in Figure 2 and Figure 3. Additionally, dilated cardiac cavities were observed, with both the right ventricle (RV) and LV measuring 30 mm in size. The ascending aorta measured 11 mm in diameter. The electrocardiogram (EKG) revealed sinus tachycardia (100 bpm), with a QRS axis of 0 degrees and a juvenile ST-T wave pattern.

Considering these findings, early-onset cardiomyopathy was identified during the course of the disease, despite the patient remaining asymptomatic. Together with the cardiologist, Holter blood pressure monitoring was conducted over a 24 h period. The average systolic pressure was noted to be 107 mmHg overall, with a daytime average of 110 mmHg and a nighttime average of 101 mmHg. The mean diastolic pressure was recorded at 64 mmHg, with consistent values of 64 mmHg during the day and 62 mmHg at night.

### 2.6. Treatment and Disease Progression

In agreement with the cardiologist, corticosteroid therapy was initiated at the age of 3 years and 8 months to provide a cardioprotective effect aimed at delaying the onset of left ventricular dysfunction, a critical complication in DMD. Prednisone was administered at a dose of 0.5 mg/kg/day under careful endocrinological monitoring of the child. Additionally, angiotensin-converting enzyme inhibitors (ACEIs) were introduced and well-tolerated, helping to maintain stable cardiac function until the age of 8. For the management of sinus tachycardia with a constant ventricular rate of 100 bpm, beta-blockers were added to the treatment regimen, achieving effective heart rate control. By the age of 9, there was a notable exacerbation of motor deficits, especially evident during stair climbing, increased incidents of falls, and difficulties in rising from the floor. Evaluations revealed an NSAA score of 20 out of 34 points and a 6 min walk test distance of 310 m. The NSAA score evolution in time is presented in Figure 4, and of the 6 min walk test in Figure 5. Despite adhering to the standards of care at that time, by the age of 11, the patient exhibited progressive deterioration in both cardiac function and muscle strength due to the underlying disease. Globally dilated echocardiographic findings revealed longitudinal dysfunction of the LV, and the ejection fraction was reduced. Persistent sinus tachycardia was noted, with a ventricular rate of 110 beats per minute. Electrocardiogram (EKG) findings indicated a QRS axis of +60 degrees, a PR interval of 0.18 s, along with minor right bundle branch block and secondary ST-T changes. At the age of 11 years and 4 months, a significant decline was observed on the NSAA scale, with a score of 7 out of 34 points and the distance covered in 6MWT having decreased to 110 m. The cardiac progression is summarized in Table 1, providing a comprehensive overview of the key findings.

## 3. Discussion

This case reflects the typical motor progression observed in DMD, with symptoms appearing at 1 year and 6 months. The patient’s motor delays and clinical features followed a pattern consistent with the natural course of the disease, leading to a diagnosis confirmed through clinical, paraclinical, and genetic evaluations [16].

In our case, a multidisciplinary approach to care was implemented, encompassing thorough monitoring across various domains as per established protocols detailed in the literature. The patient underwent comprehensive assessments in respiratory, cardiac, orthopedic, endocrinological, gastrointestinal, psychological, and psychiatric aspects [17,18]. While the patient’s respiratory function remains stable, with no need for non-invasive ventilation due to his ambulatory status [19], his cardiac assessment revealed an unusually early onset of DCM, diverging significantly from the expected disease progression reported in the literature. Endocrine parameters were within normal limits [20], while psychological and psychiatric evaluations identified disorders commonly associated with DMD [21,22].

### 3.1. Cardiac Complications

The medical literature indicates that cardiac complications in DMD manifest as a gradual onset of dilated cardiomyopathy. This condition progresses to congestive heart failure, conduction abnormalities, and ventricular or supraventricular arrhythmias, ultimately increasing the risk of sudden premature death [23,24].

Dilated cardiomyopathy can manifest at any age, although it frequently appears around 14–15 years of age and is particularly prevalent in patients older than 18 years [2,25]. Despite the progressive decline in cardiac function, DCM often remains asymptomatic for many years due to the significant reduction in energy expenditure and oxygen consumption caused by muscle weakness [26,27]. In their literature review, Dongsheng Duan et al. elaborate on the observation that cardiomyopathy typically manifests late in the progression of DMD, despite the heart being the most utilized muscle in the body. However, this case deviates from the typical pattern, with the patient demonstrating early-onset cardiac involvement, highlighting an atypical presentation that does not align with the usual progression of the disease. It is important to recognize that female carriers of DMD can face cardiac complications [28], making regular monitoring essential for early detection and timely intervention to improve outcomes.

### 3.2. Genetic and Molecular Insights

The absence or reduced levels of dystrophin, or alterations in its structure, lead to muscle membrane fragility, making muscle fibers vulnerable to damage during contraction. As the disease progresses, the muscle’s ability to repair itself diminishes, leading to necrosis of both skeletal and cardiac muscle cells, followed by their replacement with fibrofatty tissue [8]. The *DMD* gene, located on the X chromosome (Xp21), is one of the largest genes in the human genome. It spans approximately 2.4 million base pairs and consists of 79 exons. Due to its size and complexity, it is highly susceptible to mutations that can cause DMD and other dystrophinopathies [29]. The gene’s large size contributes to its increased mutation rate, with deletions being one of the most common types of pathogenic variants seen in DMD. This makes the gene particularly prone to various genetic alterations, affecting its proper function and leading to the absence or malfunction of dystrophin, a critical protein for muscle fiber stability [30]. It is expressed in a tissue-specific manner through various promoters and polyadenylation sites, producing different isoforms and splice variants [31]. Distinct promoters for brain, muscle, and Purkinje cells generate full-length dystrophin isoforms (Dp427b, Dp427m, and Dp427p) with unique first exons, which contribute to specific tissue expression patterns. The full-length dystrophin (Dp427) is predominantly found in skeletal and cardiac muscles [31,32]. According to the TREAT-NMD DMD Global Database, 86% of pathogenic variants are deletions of one or more exons, with the most common being exon 45. Large deletions tend to cluster in two regions: the distal region (exons 45–55) and the proximal region (exons 2–20) [30]. These findings support the potential for exon skipping therapy using antisense oligonucleotides (AOs) as a promising treatment for DMD [33]. Certain pathogenic variants, including those in exons 12, 14–17, 31–42, 45, and 48–49, are associated with increased cardiac involvement [34]. A meta-analysis by Zhou et al. corroborates these findings, suggesting that DMD patients with cardiac dysfunction often present with exon 45 or 46 deletions, which may serve as predictive markers for cardiac complications in these patients. These relationships are not fully understood and require further research [35].

In this case study, the patient presents with a deletion of two nucleotides in exon 55 of the *DMD* gene (Figure 1). While this variant has been previously reported in genetic databases, our literature review identified only one study mentioning the same genetic alteration. However, that study did not provide any phenotypic description, making it unclear whether a correlation exists between this specific variant and early-onset cardiac involvement in DMD. This highlights the need for further research to clarify its potential clinical implications [13].

Notably, Francesca Magri et al. [36] highlighted that, in the DMD cohort, the type of pathogenic variant did not influence clinical progression, as all patients showed a complete absence of dystrophin. However, proximal mutations were associated with earlier onset or more severe cardiac involvement in both DMD and BMD (Becker muscular dystrophy) patients, with the latter group showing a particularly higher degree of impairment.

### 3.3. Diagnostic and Therapeutic Approaches

Standard echocardiography was performed during routine cardiac evaluations, detecting longitudinal ventricular dysfunction and progressive ventricular dilation. These findings were consistent with evolving dilated cardiomyopathy, as evidenced by the increasingly globular appearance of the heart [37].

The therapeutic approach included early initiation of angiotensin-converting enzyme inhibitors to delay left ventricular dysfunction [38]. Beta-blockers were added later to manage persistent tachycardia effectively [39]. Although the patient adhered to optimal medical management, progressive deterioration in both cardiac and motor functions was observed, underscoring the challenges of managing advanced Duchenne muscular dystrophy-associated cardiomyopathy.

### 3.4. Advanced Cardiac Treatment

Complex ventricular arrhythmias, when coupled with left ventricular dysfunction or dilated cardiomyopathy, significantly increase the risk of sudden death in DMD, requiring consideration beyond pharmacological therapies, particularly in patients with heart failure [24,40]. In our current case, ongoing monitoring is essential to determine potential indications for intervention and to track the progression of cardiac comorbidity, given the current suboptimal ejection fraction that precludes invasive treatment methods. The effectiveness of implantable cardioverter-defibrillators (ICDs) in DMD remains uncertain, despite their established benefit in patients with EF less than 35% under maximal medical therapy [18]. While heart failure remains the predominant cardiac concern, the occurrence of sudden cardiac death due to arrhythmias is relatively rare in pediatric patients. Consequently, the decision to implant cardioverter-defibrillators in these patients should be carefully considered, focusing on individual patient needs and care goals. Regular cardiac monitoring and tailored use of beta-blockers for systolic dysfunction are essential components of managing cardiac health in DMD, but more research is needed to optimize these strategies and improve patient outcomes [41,42].

### 3.5. Emerging Therapies

Although existing management protocols predominantly emphasize supportive care, recent advancements in therapeutic strategies have begun targeting distinct molecular pathways implicated in DMD-related cardiomyopathy.

Vamorolone, a novel anti-inflammatory steroid analog, offers a potential alternative to traditional glucocorticoid treatment in DMD. Acting similarly to glucocorticoids but with reduced adverse effects, vamorolone may help manage inflammation while minimizing treatment-related complications [43].

Genetic therapy has emerged as a promising approach for treating DMD, aiming to restore dystrophin production [44]. Exon skipping allows the production of truncated but functional dystrophin, with PMO (Phosphorodiamidate Morpholino Oligomer)-based drugs like eteplirsen, golodirsen, viltolarsen, and casimersen demonstrating some success in skeletal muscle but limited efficacy in cardiac tissue [45,46,47]. Challenges in delivering exon-skipping therapies to the heart remain a major limitation, necessitating novel delivery strategies such as peptide-conjugated PMOs or adeno-associated virus (AAV)-mediated gene therapies [48,49].

Additionally, CRISPR/Cas9-mediated exon skipping is a developing approach with potential applications for cardiac involvement in DMD. Recent studies in mouse models have shown promising results for exon 55 skipping, restoring dystrophin levels and improving cardiac function. However, translation into clinical practice requires further refinement of delivery systems and long-term safety evaluation [50].

### 3.6. Care Coordination and Transition from Pediatric to Adult Care

Care coordination plays a vital role in this context, ensuring that care is well organized across different specialists and healthcare providers. It involves the careful integration of services, ensuring that all aspects of a patient’s condition are being addressed in a timely and comprehensive manner. For patients with DMD, care coordination includes close collaboration between pediatricians, cardiologists, neurologists, and other specialists to manage the complexities of the disease and its comorbidities. Transition planning should begin early, ideally by age 12, with formal discussions starting by ages 13–14.

These plans should be individualized, involving patients and their families to ensure they align with their needs and values [51]. The successful management of DMD and its associated cardiac conditions relies on a multidisciplinary approach [52]. The transition of care for patients with DMD who have cardiac comorbidities is a complex process requiring early and meticulous planning.

Key challenges include managing increased healthcare needs, addressing deficits in transition guidance, and ensuring coordinated care for severe cardiac conditions [53]. Care coordination helps to streamline communication among healthcare providers, ensuring no aspect of the patient’s care is overlooked and facilitating a seamless transition from pediatric to adult care. It also helps in managing follow-ups, reducing the risk of fragmented care, and ensuring that both short-term and long-term health goals are met. It is important to emphasize the need for a comprehensive transition plan for patients with DMD that addresses the entirety of their comorbidities [54]. In our case, this is particularly crucial because our patient suffers from early cardiomyopathy, which underscores the importance of early planning to manage all their comorbidities effectively, ensuring seamless continuity of care and an improved quality of life. The involvement of a coordinated team in this process ensures that all areas of health, including cardiac care, are actively monitored and treated throughout the transition period.

## 4. Conclusions

We presented the case of a DMD patient with a deletion in exon 55 of the dystrophin gene, highlighting an instance of early-onset dilated cardiomyopathy. While the significance of exon 55 deletions in early cardiac involvement remains unclear, this case underscores the need for further research into genotype–phenotype correlations to improve prognostic accuracy and guide therapy.

As life expectancy in DMD increases due to improved care, cardiac pathology has become a key determinant of survival. This report emphasizes the importance of early and consistent cardiac monitoring for timely intervention. While management remains largely supportive, ongoing research into novel therapies offers hope for improved outcomes in DMD-associated cardiomyopathy.

As the phenotype associated with this specific genetic variant has not been previously detailed, additional case reports are needed to further elucidate potential genotype–phenotype correlations.

## Figures and Tables

**Figure 1 ijms-26-01685-f001:**
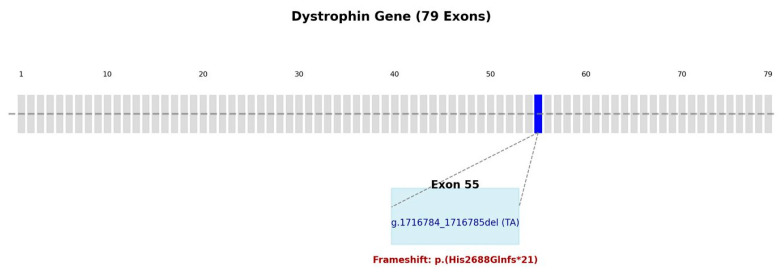
Schematic representation of the dystrophin gene (79 exons), highlighting the deletion identified in exon 55 (g.1716784_1716785delTA). This deletion causes a frameshift, leading to a premature termination codon (p.His2688Glnfs*21), resulting in the protein terminating prematurely after 21 amino acids.

**Figure 2 ijms-26-01685-f002:**
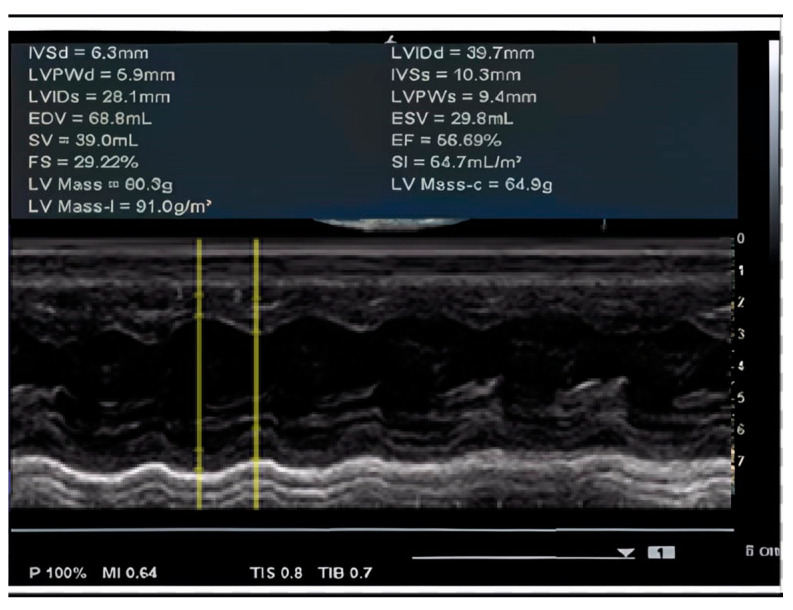
Echocardiography—evaluation of EF 56.69% by the Teichholz method.

**Figure 3 ijms-26-01685-f003:**
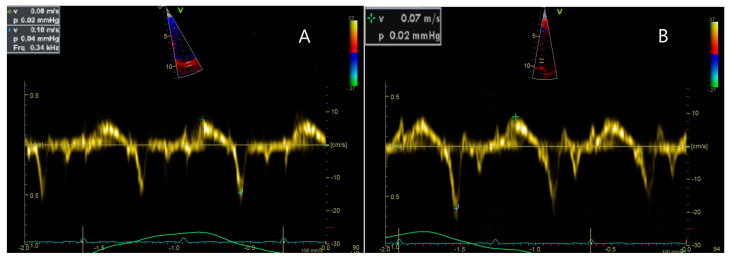
Cardiac ultrasound, incidence 4 chambers peak systolic (Sa) waves were recorded from the lateral mitral annulus (**A**, 0.8 cm/s), mitral septal annulus (**B**, 0.7 cm/s)suggestive of mild LV longitudinal dysfunction.

**Figure 4 ijms-26-01685-f004:**
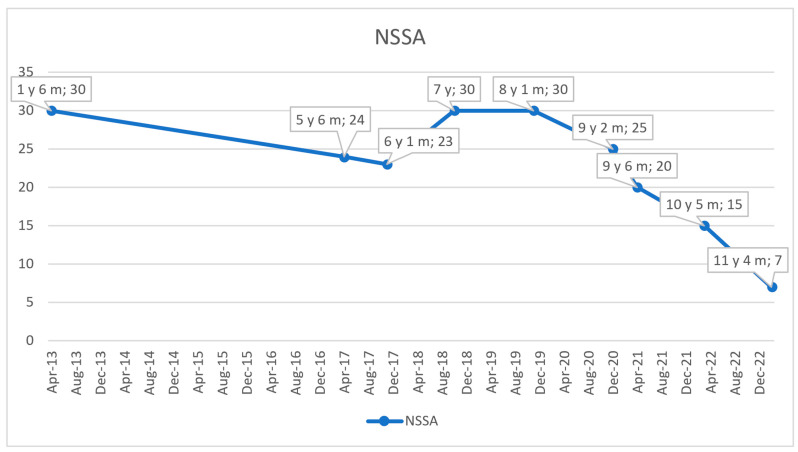
Evolution of NSAA scale scores over time; NSAA = North Star Ambulatory Assessment; y = year/s; m = month/s.

**Figure 5 ijms-26-01685-f005:**
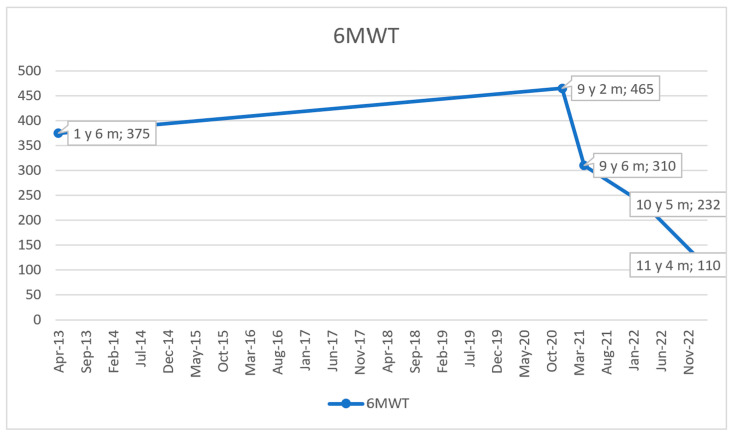
Evolution of 6MWT over time; 6MWT = 6-Minute Walk Test; y = year/s; m = month/s.

**Table 1 ijms-26-01685-t001:** Cardiac assessment over time.

Age	Cardiac Evaluation
3 y 8 m	Cardiac dimensions: 17 mm at the ascending aorta, 30 mm at both LV and RV. Doppler flow velocities of 0.90 m/s Ao and 0.90 m/s at the PV. Mitral and aortic valves appear normal with normal motility. EF = 56.69%. Dilative cardiomegaly in an incipient form in the context of DMD. EKG: Sinus tachycardia with HR = 100 bpm, QRS axis at 0 degrees, juvenile ST-T pattern.
6 y 3 m	Left ventricle dilated with normal systolic–diastolic function.
8 y 1 m	Longitudinal dysfunction of the LV and RV. EKG: Sinus tachycardia with HR = 100 bpm, QRS axis at 0 degrees, juvenile ST-T pattern.
9 y	LV slightly globular with normal global systolic function. Normal diastolic function. No valvulopathies. No pulmonary hypertension. BP = 100/60 mmHg; HR = 75 bpm; no murmurs. EKG: Sinus rhythm with HR = 90 bpm, QRS axis = +45 degrees, PR interval = 0.12 s, juvenile ST-T pattern.
10 y 5 m	LV globular. Longitudinal dysfunction of the LV. BP = 100/60 mmHg, HR = 90 bpm, grade I/VI systolic murmur heard at the left parasternal border. EKG: RBBB.
11 y 4 m	LV globular with preserved ejection fraction. Longitudinal dysfunction of the LV. EKG: Sinus tachycardia HR = 110 bpm, QRS axis +60 degrees, PR interval: 0.18 s, RBBB with secondary ST-T changes.

Ao = Aortic valve; BP = Blood pressure; DMD = Duchenne Muscular Dystrophy; EF = Ejection fraction; HR = Heart rate; LV = Left ventricle; PV = Pulmonary valve; RBBB = Right bundle branch block; RV = Right ventricle.

## Data Availability

The original data presented in this study are included in this article.

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
