# Peer review of "Early Cardiac Dysfunction in Duchenne Muscular Dystrophy: A Case Report and Literature Update"

_ijms, 2025, doi:10.3390/ijms26041685_

Round 1
Reviewer 1 Report
Comments and Suggestions for Authors
The authors presented an interesting case report with a well-documented description of a patient with Duchenne's muscular dystrophy (DMD) complicated by cardiomyopathy involvement at early stages of myocardial involvement. Despite not representing an atypical presentation or a rare scenario in patients with DMD, this manuscript brings interesting discussion for the general reader. Some points should be evaluated at this stage:
1. I suggest authors to include the classification of pathogenicity of the gene variant described in lines 74-75, according to ACMG 2015 criteria (PS4, PVS1, PM2 criteria).
2. It is important that in the start of the case description the authors describe the current age of the proband (3 years and 8 months?).
3. Was cardiac MRI performed in the reported case?
4. I suggest authors to consider the exclusion of "cardiac insufficiency" in line 181, as the same phrase describes previously "congestive heart failure".
Author Response
Dear Reviewer,
Thank you very much for your valuable insights on our paper! We appreciate the time and effort that you have dedicated to providing your feedback on our manuscript. We have been able to incorporate changes to reflect most of your suggestions. We have highlighted in yellow the changes within the manuscript.
Comment 1: I suggest authors to include the classification of pathogenicity of the gene variant described in lines 74-75, according to ACMG 2015 criteria (PS4, PVS1, PM2 criteria).
Response 1: Thank you for your suggestion. Your input has indeed added clarity to the manuscript.
We have added an introductory paragraph before the actual case presentation to provide additional context and better structure to the manuscript - “We present the case of a 19-year-old male patient with DMD associated with early-onset cardiac involvement. Clinical data, including neurological, cardiac, and functional assessments, as well as genetic testing results, were retrospectively collected and analyzed. Cardiac function was evaluated through routine echocardiographic and electrocardiographic monitoring. The pathogenic variant was identified using NGS tests in the peripheral blood and classified as pathogenic using American College of Medical Genetics criteria (13).”
We have also revised the “Genetic Testing” section to include more specific details, which now reads as follows: “We performed phenotype-drive genetic testing. The single gene filtered NGS showed a pathogenic (PVS1, PM2, PS4) variant in dystrophin gene, NM_004006.3(DMD): exon 55 (g.1716784_1716785del; c.8064_8065delTA;p.(His2688GlnfsTer21) confirming the diagnosis of DMD at the age of three. “Additionally, Figure 1 now illustrates this genetic finding for clarity.

Figure 1. Schematic representation of the dystrophin gene (79 exons), highlighting the deletion
identified in exon 55 (g.1716784_1716785delTA). This deletion causes a frameshift, leading to a
premature termination codon (p.His2688Glnfs*21), resulting in the protein terminating
prematurely after 21 amino acids.”
Comment 2: It is important that in the start of the case description the authors describe the current age of the proband (3 years and 8 months?).
Response 2: Thank you for your comment. We have now revised the text to include that the proband is 19 years old. This information can be found in lines 67–68 - “We present the case of a 19-year-old male patient with DMD associated with early-onset cardiac involvement.”
Comment 3: Was cardiac MRI performed in the reported case?
Response 3: Thank you for raising this point. Unfortunately, cardiac MRI is not reimbursed by the national healthcare system in Romania, limiting its accessibility for our patient. Initially, standard echocardiography was performed as part of routine cardiac evaluations. More recently, we have adopted speckle tracking echocardiography, a technique highly regarded for its ability to detect early left ventricular dysfunction, with a diagnostic performance closely approximating that of cardiac MRI.
Comment 4: I suggest authors to consider the exclusion of "cardiac insufficiency" in line 181, as the same phrase describes previously "congestive heart failure".
Response 4: We located "cardiac insufficiency" it in line 195 and have removed it
- “This condition progresses to congestive heart failure, conduction abnormalities, and ventricular or supraventricular arrhythmias, ultimately increasing the risk of sudden premature death.”

Reviewer 2 Report
Comments and Suggestions for Authors
This ms reports a case of Duchenne Muscular Dystrophy, but its relevance to the biomedical community interested in such disease remains unclear.
_No genetic data are shown and the methodological section is sketchy to say the least (see below the “Genetic testing” text from the ms, dystrophin is not even mentioned?). A scheme supporting them and helping the reader to better understand in which position the deleted nucleotides are located is also missing.
“2.2. Genetic testing
Following clinical and paraclinical evaluations that raised suspicion of a neuromus- cular disorder, the decision was made to pursue genetic testing for a definitive diagnosis. Genetic testing revealed a deletion of two nucleotides in exon 55 (g.1716784_1716785del; c.8064_8065delTA; p.(His2688iNFS*21)), confirming the diagnosis of DMD at the age of three.”
_It is unclear/not mentioned whether the deletion found is novel or not. The authors, rather enigmatically, state “a rare variant with limited literature” (?).
_I did not find any reference to Figure 2 in the ms text.
_I believe that some important references are missing, as a notable example see the outstanding work by Rok and colleagues (Prevention of early-onset cardiomyopathy in Dmd exon 52–54 deletion mice by CRISPR-Cas9-mediated exon skipping, Molecular Therapy, 2023) showing that skipping of exon 55 ameliorates significantly the cardiac phenotype in model mice.
Author Response
Dear Reviewer,
Thank you for your constructive feedback on our submission. We have carefully addressed each of your suggestions, and the following responses outline the revisions made to the manuscript.
Comment 1: No genetic data are shown and the methodological section is sketchy to say the least (see below the “Genetic testing” text from the ms, dystrophin is not even mentioned?). A scheme supporting them and helping the reader to better understand in which position the deleted nucleotides are located is also missing.
Response 1: Thank you for your observation. We have added an introductory paragraph to provide the necessary context and details, which now reads: “We present the case of a 19-year-old male patient with DMD associated with early-onset cardiac involvement. Clinical data, including neurological, cardiac, and functional assessments, as well as genetic testing results, were retrospectively collected and analyzed. Cardiac function was evaluated through routine echocardiographic and electrocardiographic monitoring. The pathogenic variant was identified using NGS tests in the peripheral blood and classified as pathogenic using American College of Medical Genetics criteria.”
Furthermore, we have revised the “Genetic Testing” section to include more specific details, which can now be found between lines 83–87 - “We performed phenotype-drive genetic testing. The single gene filtered NGS showed a pathogenic (PVS1, PM2, PS4) variant in dystrophin gene, NM_004006.3(DMD): exon 55 (g.1716784_1716785del; c.8064_8065delTA;p.(His2688GlnfsTer21) confirming the diagnosis of DMD at the age of three. We have illustrated this genetic finding in Figure 1.”
Additionally, we have created a scheme illustrating the position of the deleted nucleotides to help the reader better understand the genetic findings, following your recommendation.
Figure 1. Schematic representation of the dystrophin gene (79 exons), highlighting the deletion identified in exon 55 (g.1716784_1716785delTA), which causes a frameshift mutation (p.His2688Glnfs*21).

Comment 2:
“2.2. Genetic testing
Following clinical and paraclinical evaluations that raised suspicion of a neuromus- cular disorder, the decision was made to pursue genetic testing for a definitive diagnosis. Genetic testing revealed a deletion of two nucleotides in exon 55 (g.1716784_1716785del; c.8064_8065delTA; p.(His2688iNFS*21)), confirming the diagnosis of DMD at the age of three.”
_It is unclear/not mentioned whether the deletion found is novel or not. The authors, rather enigmatically, state “a rare variant with limited literature” (?).
Response 2: We were unable to determine whether the variant is de novo or inherited, as the patient’s mother was not tested due to personal reasons.
We have revised the sentence as suggested, and it now reads as follows: “In this case study, the patient presents with a deletion of two nucleotides in exon 55 of the dystrophin gene (Figure 1). While this variant has been previously reported in genetic databases, our literature review did not identify data specifically linking this genetic deletion to early-onset cardiac involvement in patients with DMD. This underscores the need for further research to better understand its potential clinical implications.”
Comment 3:_I did not find any reference to Figure 2 in the ms text.
Response 3: Thank you for your observation. We have revised the paragraph accordingly and included the figure in the text- “Despite the absence of symptoms, the echocardiographic examination confirmed longitudinal dysfunction of the left ventricle with an ejection fraction of 56.69%, along with evidence of mild ventricular dilation, as seen in Figure 2 and Figure 3.” Please note that the figure number has been updated, as we inserted a new Figure 1 earlier in the manuscript.
Comment 4:_I believe that some important references are missing, as a notable example see the outstanding work by Rok and colleagues (Prevention of early-onset cardiomyopathy in Dmd exon 52–54 deletion mice by CRISPR-Cas9-mediated exon skipping, Molecular Therapy, 2023) showing that skipping of exon 55 ameliorates significantly the cardiac phenotype in model mice.
Response 4:
Thank you for highlighting this key study. We have included the reference and integrated its findings into the discussion:
- “Among the genetic therapies, CRISPR/Cas9-mediated exon skipping targeting exon 55 has shown promise for addressing cardiac complications in DMD. A study conducted by Rok et al.(58) using the Dmd Δ52–54 mouse model demonstrated that a single-guide RNA approach targeting the splice donor site was the most effective in restoring dystrophin levels in the heart and preventing early cardiac dysfunction. By increasing the sgRNA dosage, the researchers enhanced editing efficiency in cardiac tissue, though further improvements in protein expression are still needed. These findings in mouse models highlight the potential of exon 55 skipping to mitigate cardiomyopathy in DMD and emphasize the need for refining delivery systems and sgRNA design to translate these results into clinical applications.”
